# Reducing Manual Annotation Costs for Cell Segmentation by Upgrading Low-Quality Annotations [note 1]

**DOI:** 10.3390/jimaging10070172

**Published:** 2024-07-17

**Authors:** Serban Vădineanu, Daniël M. Pelt, Oleh Dzyubachyk, Kees Joost Batenburg

**Affiliations:** 1Leiden Institute of Advanced Computer Science, Leiden University, 2311 EZ Leiden, The Netherlands; d.m.pelt@liacs.leidenuniv.nl (D.M.P.); k.j.batenburg@liacs.leidenuniv.nl (K.J.B.); 2The Division of Image Processing, Leiden University Medical Center, 2333 ZA Leiden, The Netherlands; o.dzyubachyk@lumc.nl

**Keywords:** annotation enhancement, annotation errors, cell segmentation, deep learning

## Abstract

Deep-learning algorithms for cell segmentation typically require large data sets with high-quality annotations to be trained with. However, the annotation cost for obtaining such sets may prove to be prohibitively expensive. Our work aims to reduce the time necessary to create high-quality annotations of cell images by using a relatively small well-annotated data set for training a convolutional neural network to upgrade lower-quality annotations, produced at lower annotation costs. We investigate the performance of our solution when upgrading the annotation quality for labels affected by three types of annotation error: omission, inclusion, and bias. We observe that our method can upgrade annotations affected by high error levels from 0.3 to 0.9 Dice similarity with the ground-truth annotations. We also show that a relatively small well-annotated set enlarged with samples with upgraded annotations can be used to train better-performing cell segmentation networks compared to training only on the well-annotated set. Moreover, we present a use case where our solution can be successfully employed to increase the quality of the predictions of a segmentation network trained on just 10 annotated samples.

## 1. Introduction

Deep-learning algorithms have been providing effective solutions for many tasks, contributing to the advancement of domains such as speech recognition [1] and computer vision [2]. One important computer vision task that is being tackled with such algorithms is image segmentation [3], a process that labels each pixel into categories, e.g., background and various cell types. However, deep-learning models require large quantities of annotated data for training. In addition, the provided annotations should also be of high quality. Specifically, the annotations should be accurate by providing information that reflects the reality within the input, and be complete, meaning that they provide all the information required for the given task, e.g., all pixels from an image have an associated label in a segmentation task.

For many biomedical imaging tasks, including cell imaging [4], the annotations are created manually by domain experts. Due to the limited availability of experts, the annotation process is often tedious [5], limiting the capacity for annotating the large quantities of data required by deep-learning algorithms. As a result, the general adoption of deep learning for such specialized domains may be considerably hindered. An annotation process with fewer quality constraints could significantly reduce the burden on expert annotators, enabling them to produce annotated images within a shorter time frame. For instance, when creating segmentation masks, the boundary of every cell in the image has to be carefully delineated. By providing coarser delineations, only annotating a subset of all cells, or relying on automatic but inaccurate segmentation tools based on classical image processing, a much faster annotation process can be achieved. However, training directly on low-quality annotations harms the performance of cell segmentation deep-learning algorithms [6]. Thus, it becomes apparent that a solution that leverages inaccurate annotations to expand costly training data sets can greatly benefit the adoption of deep learning for cell segmentation.

Learning from imperfect or missing labels due to annotation constraints is a long-standing issue associated with machine-learning tasks. In the case of cell segmentation, obtaining large amounts of labeled data requires time-consuming efforts by experts with specialized knowledge of the task. One field concerned with this problem is weakly-supervised learning, where the aim is to train deep-learning algorithms to produce complete segmentation masks by only providing the models with partial annotations. Such techniques usually vary in the amount of information that is present in the annotations, which can include bounding boxes [7], rough sketches of shape contours [8], geometrical shape descriptors in the form of center points and lines [9], or partially-annotated segmentation areas [10]. Despite their promising results, these techniques are generally tailored towards a single type of inconsistency, which can limit their applicability.

Directly accounting for labelling errors, implicit consistency correction methods compensate for inaccuracies in the annotated input during the training process by, for instance, reducing the influence of gradients coming from segmentation areas of lower confidence [11], by using a teacher–student architecture [12] to change the label of less confident areas in the annotation mask [13], or by using adversarial training to only annotate high-confidence areas of unlabeled data [14]. On the other hand, explicit consistency correction solutions provide fine adjustments to the output of trained deep-learning models [15,16,17,18]. Similarly to weakly supervised techniques, these methods lack a broad applicability and their utilization depends rigidly on custom architectures. When it comes to improving the provided labels, Yang et al. [19] developed a solution for iteratively adjusting the manual annotations of retinal vessels by employing generative adversarial networks. Their framework, however, only produces small adjustments, relies on a relatively large amount of high-quality annotations, and may suffer from the challenges associated with generative models, e.g., mode collapse and convergence failure [20].

Also concerned with annotation scarcity, few-shot segmentation aims to segment new query images by leveraging information from relatively few support images with a limited amount of annotations. However, these approaches generally require additional training tasks with a large set of semantic classes [21,22] whose annotations can be costly to obtain. The need for manual annotations can also be avoided by employing general foundation models such as the Segment Anything Model (SAM) [23], or cell-specific models such as Cellpose [24]. However, although the applicability of such models is not confined to a single image modality or cell type, they do not generalize well to images outside their vast training pool. For instance, the SAM is not accurate when the targets have weak boundaries [25], which can be the case with cell images [26], whereas Cellpose is sensitive to variations in the texture of the objects [24]. This may make these general solutions less suitable than techniques trained for a specific cell type.

In summary, although there are many methods designed for improving deep-learning segmentation with incorrect or incomplete labels, these solutions generally tackle specific types of inconsistencies, e.g., boundary uncertainty, require custom architectures or training schemes, or have considerable annotation requirements for additional training tasks. In this paper, we present a method designed to be applied to a wide set of inconsistencies, with low data requirements and a flexible training scheme allowing for a straightforward integration in other pipelines. We propose a framework for effectively obtaining large amounts of high-quality training data with limited required human annotation time for the task of cell segmentation. Our approach is based on manually annotating a small training set of high quality, which we then enlarge with a much larger set with low-quality annotations (possibly produced with considerably less human effort). In order to leverage the low-quality annotations, we train a convolutional neural network to learn the mapping for upgrading a low-quality annotation to a high-quality one by presenting it with both high-quality annotations as well as low-quality versions of these annotations. We create multiple types of erroneous annotations by perturbing the high-quality annotations with a function that approximates potential errors resulting from a low-quality annotation process. Moreover, we show that this perturbation function does not need to exactly replicate the annotation errors present in the low-quality annotations in order for a good mapping to be trained. The training process requires pairs of the perturbed annotations with their corresponding images as input for the upgrade network with the unperturbed, high-quality annotations as targets. We apply the learned mapping to the large low-quality set to enhance its annotations. Finally, we combine the initial small set of well-annotated data together with the larger set with upgraded annotations and use them for training accurate deep-learning models for the task of cell segmentation. By separating the inconsistency correction step, i.e., the upgrading of annotations, from the segmentation step, we enable our framework to tackle a wide array of inconsistencies and we facilitate its integration into other segmentation pipelines.

This paper is an extended version of our paper published in [27]. We extend our previous version by:Providing a formal introduction of the perturbations used throughout the experiments together with illustrative examples;Adding extensive experiments to further evaluate (i) the training requirements of the upgrade network, (ii) the effect of inconsistencies in the annotations of the high-quality data set, (iii) the trade-off between annotation cost and segmentation performance when upgrading the low-quality annotations, (iv) the performance of segmentation networks based on the annotation quality;Adding comparisons with related solutions;Adding experiments on complex RGB cell segmentation data sets;Expanding the discussion by elaborating on the potential consequences of our observations and scenarios under which our framework can contribute the most.

## 2. Materials and Methods

### 2.1. Data Sets

#### 2.1.1. Synthetic Data

We opted to use synthetic data to study most aspects of our method since their ground truth annotations do not suffer from the inconsistencies a human annotator may induce. Thus, we can be confident that such external factors do not influence the outcomes of our experiments. Also, to isolate the effect of a particular type of inconsistency in the low-quality set, we apply perturbation functions (see Section 2.2.3) throughout the experimentation with synthetic data. We employ three data sets [28], which consist of microscopy images of HL60 nuclei cells, granulocytes, and both cell types, respectively, produced by a virtual microscope [29] extracted from the Masaryk University Cell Image Collection (https://cbia.fi.muni.cz/datasets) (accessed on 15 June 2023). Each data set consists of 30 volumes of 129 slices, each containing 565×807 16-bit pixels. We filtered the volumes by eliminating the slices with empty labels, which resulted in differently sized volumes, averaging 84 slices per volume. In addition, 25 volumes were used for training, while 5 volumes were kept only for testing. In Figure 1, we show sample slices and their corresponding high-quality annotations of the synthetic data sets. Since they are organized in volumes, we want to avoid selecting high-quality annotations of adjacent slices since such samples show little variation in their input and may be less informative when training the upgrade network than more distant slices. Consequently, we sample by subdividing a volume into a number of sections equal to the number of slices we want to select. We then select the middle slice of each section, thus ensuring an equidistant separation between slices. Additionally, when we sample from multiple volumes, we similarly partition each volume, but we select every next slice from a section belonging to a different volume in a circular manner. For instance, when taking a total of 5 slices from 5 volumes, the first slice will be selected from the center of section 1 from volume 1, the second from section 2 of volume 2 and so on.

#### 2.1.2. Real Data

We also employ two manually-annotated data sets: the EPFL Hippocampus data set [30] and a large-scale data set for colonic nuclear segmentation called Lizard [31]. The EPFL data set is comprised of a training and a testing volume, each containing 165 slices of 768 × 1024 8-bit grayscale pixels. This set of images, obtained using focused ion beam scanning electron microscopy, is commonly used for benchmarking mitochondria segmentation algorithms, whose monitoring can provide, for instance, insights into the development of neurodegenerative diseases [32]. The Lizard data set contains histology RGB images of colon tissue of varying sizes with instance labels for each cell. Among the six cell types annotated in this data set, we selected the most prevalent category, i.e., epithelial cells, as our target and the remaining cells as background objects. This choice allows us to test our method on the largest number of samples, which ensures that we obtain the most statistically significant results. We split the images into 500×500 patches, with 100 pixels overlapping between patches and removed patches that did not contain epithelial cells. We partitioned the resulting set into 1209 training and 288 testing patches. In this case, we assume the corresponding provided ground-truth annotations to be of high quality. For each data set, we select a small subset of samples for which we keep the high-quality annotations while perturbing the annotations of the remaining samples to generate the low-quality set. This perturbation step is performed only once per annotated image.

### 2.2. Method

In Figure 2, we illustrate an overview of our method. We consider a high-quality annotation process that produces labels in a slow and costly manner and a low-quality annotation process, yielding labels faster and cheaper. Within a given time frame, the processes would generate a small data set with high-quality labels and a larger lower-quality set. We apply perturbations to the well-annotated labels and we use the perturbed labels together with their corresponding images as input to train an upgrade model. We employ the upgrade model to enhance the labels of the larger data set, which we use in conjunction with the well-annotated samples to train the final segmentation model.

#### 2.2.1. Background

We apply our framework to the segmentation task of 2-dimensional vector-valued (e.g., RGB, grayscale) images. In this paper, we define an image as a matrix of pixels $x\in R^{N\times M\times C}$, where *N*, *M*, and *C* represent the number of rows, columns, and channels, respectively. The goal of segmentation is to create a mapping from a given input *x* to the target $y\in Z^{N\times M}$ in order to provide a separation between the different entities within that image. Essentially, a label is attributed to each pixel according to the entity that it belongs to. When using deep learning for image segmentation, this mapping is approximated using convolutional neural networks (CNNs), $f_{\delta }:R^{N\times M\times C}\to R^{N\times M}$, which require a set of image-target pairs $X=\{\left(x_{1},y_{1}\right),\left(x_{2},y_{2}\right),\ldots ,\left(x_{N_{t}},y_{N_{t}}\right)\}$, to train their parameters, δ. The process of training neural networks usually involves successive predictions based on the input *x* and adjusting the parameters such that the loss between the predictions and the labels is minimized. In order to achieve the desired results, the network requires well-annotated training samples. We describe the annotation process that produced high-quality labels as the output of the high-quality annotator,
(1)$$A^{HQ}:R^{N\times M\times C}\to A^{HQ},$$
that receives an input image *x* and produces a label that belongs to the set of high-quality annotations, $A^{HQ}$, i.e., it is both complete and correct. Such an annotation can be the result of a consensus between multiple experts or can require a slow and careful delineation of the shape of each element in *x* by a single expert. Additionally, we define the set of well-annotated images, $X^{HQ}=\left\{\left(x,A^{HQ}\left(x\right)\right)\right\}$, needed to train the network parameters,
(2)$$\hat{\delta }=\underset{\delta }{\mathrm{argmin}}\sum_{\left(x,y\right)\in X^{HQ}}L\left(f_{\delta }\left(x\right),y\right),$$
where *L* is a loss function. Due to their large parameter count, these models are generally prone to overfitting and therefore require large quantities of well-annotated samples.

#### 2.2.2. Perturbation-Removal Framework

Since producing a sufficient number of high-quality annotations may prove unfeasible for cell segmentation, the required annotations may be supplied via a less rigorous annotation process. A low-quality annotation process would, for instance, result from an individual expert who quickly produces the annotation, without spending additional time on finer shape details or on removing ambiguities. Also, for setups that require consensus, the label can come from a single expert, a person in training, or a non-expert, thus reducing the annotation costs. Alternatively, the low-quality annotations can even be the product of traditional segmentation techniques (e.g., thresholding, graph cut [33], Otsu [34]) or machine-learning-based algorithms, removing the need for a human annotator in this stage of the process. For instance, one easily-applicable strategy to produce low-quality annotations is to simply train a segmentation network on the few available high-quality samples and then use its predictions on the remaining unannotated samples as low-quality annotations. We define the low-quality annotator
(3)$$A^{LQ}:R^{N\times M\times C}\to A^{LQ},$$
as a function that produces labels that are either incorrect or incomplete or both, thus, being included in the set of low-quality annotations, $A^{LQ}$.

Training solely with low-quality annotations generally leads to inaccurate results [6]. Thus, we propose a solution to enhance the quality of a larger set of low-quality annotations, which we utilize to enlarge an initially small set of high-quality annotations. Our framework requires a small number of well-annotated images, $X^{HQ}$, together with a substantially larger set of images and their low-quality annotations, $X^{LQ}=\left\{\left(x,A^{LQ}\left(x\right)\right)\right\}$, with $\left|X^{HQ}|<|X^{LQ}\right|$. We aim to enhance $A^{LQ}\left(x\right)$ to $A^{HQ}$ by finding the upgrade function
(4)$$U:\left(R^{N\times M\times C},A^{LQ}\right)\to A^{HQ},$$
which translates an annotation of the input image created by the low-quality annotator to an annotation belonging to the space of high-quality annotations. In order to create both high- and low-quality versions of annotations, we utilize a perturbation function that aims to approximate the unknown mapping from a high-quality annotation to a low-quality one. We handcraft function
(5)$$P:A^{HQ}\to A^{LQ},$$
which applies perturbations to a high-quality annotation to create an annotated image that approximates a faster, but lower-quality, annotation process. The choice for such a function can vary by task and data set, with implementations that can include heuristics or even learning the perturbations from the data. In our work, we assume that we can approximate the perturbation function, *P*, by implementing a custom stochastic version of it. Additionally, we assume that the function *U* that maps the low-quality label to a high-quality one is a learnable function. We employ the high-quality set to generate many $\left(x,P\left(A^{HQ}\left(x\right)\right)\right)$ pairs. Given the stochastic nature of our chosen perturbation function, we can generate multiple perturbed versions of the same high-quality annotation; thus, we only require a small number of $\left(x,A^{HQ}\left(x\right)\right)$ pairs. We utilize the generated pairs to train an upgrade network, $u_{\theta }$, parameterized by θ, which approximates *U* by finding
(6)$$\hat{\theta }=\underset{\theta }{\mathrm{argmin}}\sum_{\left(x,A^{HQ}\left(x\right)\right)\in X^{HQ}}L\left(u_{\theta }\left(x,P\left(A^{HQ}\left(x\right)\right)\right),A^{HQ}\left(x\right)\right),$$
where *L* is a loss function. After training $u_{\theta }$, we apply it to our lower-quality set. In this way, we enhance the low-quality annotations, which results in the pairs $\left(x,u_{\theta }\left(x,A^{LQ}\left(x\right)\right)\right)$ of input images and upgraded annotations. Finally, we use both the enhanced $\left(x,u_{\theta }\left(x,A^{LQ}\left(x\right)\right)\right)$ pairs and the initial high-quality $\left(x,A^{HQ}\left(x\right)\right)$ pairs as training samples for our final segmentation task. Therefore, our segmentation CNN $f_{\delta }$ will be obtained as
(7)$$\begin{array}{c} \hat{\delta }=\underset{\delta }{\mathrm{argmin}}(\sum_{\left(x,y\right)\in X^{HQ}}L\left(f_{\delta }\left(x\right),y\right)+\\ \sum_{\left(x,y\right)\in X^{LQ}}L\left(f_{\delta }\left(x\right),u_{\hat{\theta }}\left(x,y\right)\right)). \end{array}$$

Algorithm 1 shows the pseudocode of a segmentation pipeline that makes use of our upgrade network. The requirements of our framework are (1) a small set with high-quality annotations, (2) a larger set with low-quality annotations, and (3) a perturbation function. The objective of this pipeline is to obtain the parameters δ of a well-trained segmentation network. We initially train the upgrade network $u_{\theta }$ only on the high-quality data $X^{HQ}$, whose labels we perturb with the previously selected perturbation function, *P*. We aim here to obtain predictions from input images and perturbed labels that match the high-quality annotations as closely as possible. After estimating the parameters of $u_{\theta }$, we apply it to $X^{LQ}$, whose images and resulting upgraded annotations we employ, in conjunction with $X^{HQ}$, to estimate the parameters δ of a segmentation network.
**Algorithm 1** Upgrade Framework**Require:** $X^{HQ},X^{LQ},P$ **return** δ
 (1) Train the upgrade network $u_{\theta }$:
 **for** $\left(x,y\right)\in X^{HQ}$ **do**
    Perturb *y*: P(y)
    Predict upgraded label: $u_{\theta }\left(x,P\left(y\right)\right)$
    Compute loss: $L(u_{\theta }\left(x,P\left(y\right)\right),y)$
 **end for**
 Estimate θ^ according to Equation (Equation 6)
 (2) Upgrade low-quality set and expand segmentation training data:
 **for** $\left(x,y\right)\in X^{LQ}$ **do**
    Upgrade low-quality label: $u_{\theta }\left(x,y\right)$
 **end for**
 Estimate $\hat{\delta }$ according to Equation (Equation 7)


#### 2.2.3. Producing Low-Quality Annotations

We designed our method for the task of binary cell segmentation, where the object of interest is a single type of cell. In order to apply our perturbation function, we require the instance label of every cell in the image. Therefore, considering *E* cells in image *x*, we define L⊂Z as the set of all cell instance labels, with |L|=E. Our label then becomes
(8)$$\begin{array}{c} y_{nm}=\left\{\begin{array}{cc} i, & \mathrm{if}\;x_{nm}\;\mathrm{belongs}\;\mathrm{to}\;\mathrm{cell}\;i\in L,\\ 0, & \mathrm{otherwise} \end{array}\right.\;\\ \;1\leq n\leq N,\;1\leq m\leq M. \end{array}$$

We apply three types of perturbations (omission, inclusion, and bias), introduced in [6], which are designed to reflect the incompleteness and inaccuracy of the cell segmentation masks resulting from an annotation process with fewer resources. For instance, a much shorter annotation time can be spent by using segmentation masks that only contain a proportion of the total cells present in the image. Moreover, allowing for inconsistencies in cell recognition in the form of inclusions can also reduce the time an annotator spends choosing which cells to include in the segmentation mask. Finally, by eliminating the need to provide correct cell border delineations, we can expect a boost in the annotation speed.

**Omission Perturbation**. We randomly select a subset of S≤E of cell instance labels $L^{S}\subseteq L$, whose size is chosen such that it satisfies the *omission rate* $r_{\omega }=\frac{S}{E}$. Our perturbation function, therefore, becomes
(9)$$\begin{array}{c} P{\left(y\right)}_{nm}=\left\{\begin{array}{cc} 0, & \mathrm{if}\;x_{nm}\;\mathrm{belongs}\;\mathrm{to}\;\mathrm{cell}\;i\in L^{S},\\ y_{nm}, & \mathrm{otherwise} \end{array}\right.\;\\ \;1\leq n\leq N,\;1\leq m\leq M. \end{array}$$**Inclusion Perturbation**. Given an image *x* and Λ⊂Z, a set of instance labels of other objects belonging to *x* (L∩Λ=∅), we perform inclusion by randomly selecting a subset $\Lambda^{S}\subseteq \Lambda$ of the objects, whose size S≤F satisfies the *inclusion rate* $r_{\phi }=\frac{S}{F}$. Hence, we apply the perturbation as
(10)$$\begin{array}{c} P{\left(y\right)}_{nm}=\left\{\begin{array}{cc} j, & \mathrm{if}\;x_{nm}\;\mathrm{belongs}\;\mathrm{to}\;\mathrm{shape}\;j\in \Lambda^{S},\\ y_{nm}, & \mathrm{otherwise} \end{array}\right.\;\\ \;1\leq n\leq N,\;1\leq m\leq M. \end{array}$$**Bias Perturbation**. We model the inconsistency in border delineation by performing morphological operations [35] on the cell labels. We employ dilation operations, *D*, to enlarge the cell area and erosion operations, *E*, to shrink the cell area. The operation is randomly chosen and the impact of the operation is controlled by factor *q* that controls the number of iterations, with a 3×3 all-ones matrix as the fixed structural element, for which we perform the chosen operation. This bias severity constant, randomly picked between 1 and $q_{max}$, indicates the largest allowed number of iterations. As a result, the perturbation is formed either as
(11)$$P\left(y\right)=E^{q}\left(y\right)\;\mathrm{or}\;P\left(y\right)=D^{q}\left(y\right).$$
where $E^{q}$ and $D^{q}$ denote *q* iterations of erosion and dilation, respectively.

Given the relatively ill-defined distinction between low-quality and high-quality annotations, we will further consider as low-quality annotations only the ones affected by large degrees of perturbations, i.e., 70% omission, 70% inclusion, a bias of 6, or a combination of perturbations. Thus, we only consider as low-quality the annotations that significantly diverge from the gold standard. In Figure 3, we illustrate an example of an annotation where all three perturbation types are present and highlighted. Alternatively, we investigate the case where the low-quality annotator $A^{LQ}$ would not imply any human effort. This can happen when $A^{LQ}$ are produced by a segmentation network trained on the small number of samples in the high-quality set $A^{HQ}$. In this case, the generation of low-quality annotations is disentangled from the perturbations that we apply when training the upgrade network.

### 2.3. Experimental Setup

We designed our experimental setup around a PyTorch [36] implementation of UNet [37]. UNet features an encoder–decoder architecture with skip connections between the encoding layers and the decoding layers of the same spatial resolution. We employed 4 convolutional blocks in the encoder and 4 in the decoder, with a block containing 2 convolutional and 2 batch normalisation layers. We treat both the segmentation and upgrade tasks as binary pixel-wise classification tasks. Thus, the output of the network in both cases is a two-channel image with the first channel’s pixels being 0 if they belong to the foreground and 1 if they belong to the background, with the opposite holding true for the second channel. All activations between layers are ReLU functions, with the exception of the last layer, where the output is processed by a soft-max function. We train the network until there is no improvement in the validation score for 10 consecutive epochs, at which point we only keep the model with the highest score. Our loss function is the Dice loss, and we update the network’s parameters according to ADAM optimization algorithm [38], with a learning rate of $10^{-5}$ and a batch size of 4. We partition our data into training and testing with an additional 80/20 split of the training data into training and validation. Finally, we present our results by reporting the Sørensen–Dice coefficient computed over the entire test set and averaged over 5 runs. We validated our comparisons by using the Wilcoxon non-parametric test [39].

## 3. Results

We performed a series of experiments to analyze various aspects of our proposed framework. In Section 3.1, we use the synthetic data sets with objective ground truth to measure the quality gain of upgraded annotations under various sets of assumptions. On the same data sets, we also evaluate the benefits of expanding the segmentation training data with upgraded annotations in terms of segmentation performance and annotation cost (Section 3.2). Furthermore, in Section 3.3, we validate our previous observations on real manually-annotated data. Lastly, we show, in Section 3.4, a case study of an application where our solution can be integrated to improve the prediction quality of a segmentation network trained with insufficient samples.

### 3.1. Analysis of the Upgrade Network

To assess the optimal training set size for the upgrade network $u_{\theta }$, we created various training sets by varying both the total number of annotated slices and the number of volumes from which the annotated slices were selected. The models were trained to upgrade annotations affected by 70% omission, 70% inclusion, and a bias of 6, respectively. The results presented in Figure 4 show that the upgrade network requires just 5 well-annotated slices to improve the quality of the annotations, regardless of the applied perturbation. We also notice that the resulting quality of the upgraded annotations plateaus quickly to Dice values > 0.9. We report the optimal number of training slices for different perturbations together with the corresponding Dice score of the upgraded annotations in Table 1.

So far, we assumed that we can perfectly model the errors affecting the low-quality annotations with our perturbation functions. However, in practice, it might be difficult to exactly match the type and severity of the perturbations present in the data. To account for that, we relax this assumption by allowing a mismatch between the error generated by the perturbation functions and the errors in $X^{LQ}$. In Table 2, we report the effect of such mismatch on the performance of the upgrade network when the annotations of $X^{LQ}$ contain 30% omission, 30% inclusion, and a bias severity of 4, respectively. We observe that, even when not reaching the highest Dice scores, the upgraded annotations show high Dice scores when $u_{\theta }$ is trained on the highest perturbation level. This implies that varying the presence of a large proportion of the cell masks can be more beneficial for training $u_{\theta }$ than aiming to exactly match the amount of error present in the $X^{LQ}$.

In addition to the perturbation function, another essential requirement of our solution is the presence of a high-quality set of annotations for training $u_{\theta }$. Since we use synthetic data, the quality of this set is ideal, which, however, is not expected from manual annotations for many reasons, including inter-observer variability [40] or limited available resources. We model these inaccuracies by introducing moderate amounts of perturbations into the high-quality set. Figure 5 illustrates that the upgrade networks trained on the larger HL60 cells are robust to imperfect HQ annotations, whereas the ones trained on granulocytes are more sensitive due to the comparatively smaller footprint of the cells. Thus, the same amount of perturbation affects the quality of the granulocytes annotations more drastically than that of HL60 cells. Despite allowing for a moderate amount of omission and inclusion perturbations, the networks trained on granulocytes show a sharp drop in performance for bias since this type of perturbation introduces the greatest variation in shape relative to cell size among the two data sets.

We compare our solutions with works tackling the issue of training biomedical image segmentation models with imperfect or incomplete annotations. We selected techniques that employ full-size segmentation masks for training and that apply corrections to these masks to either fill incomplete areas or remove incorrect ones. Also, although we compare the selected methods for all our perturbation types, it is important to note that Partial Labeling [10] was designed for setups closer to omission than the other perturbations, whereas Confident Learning [13] tackles uncertain areas at the border of the masked areas resembling more our bias perturbation. In Table 3, we observe that our method generates comparable results with Partial Labeling for omission and Confident Learning for bias perturbation. However, among all three perturbation types, our framework performs consistently better than the other solutions, showing wider applicability to different types of inconsistency.

### 3.2. Segmentation Improvements

In Section 3.1, we investigated the capability of the upgrade network to improve the quality of annotations affected by errors. In this section, we are analyzing whether adding the upgraded annotations to the training set results in improved segmentation performance and reduced overall annotation costs. In Table 1, we report different scenarios under which $X^{HQ}$ and $X^{LQ}$ can be used to train networks for segmentation. Given an initial data set with low-quality annotations, we can use it directly as a training set for segmentation (LQ only column in Table 1). We can also spend additional resources on improving the quality of a small number of annotations and utilize them in conjunction with the low-quality set (column HQ + LQ) or we can employ the high-quality set alone for training (column HQ). Finally, we can use our framework for upgrading the low-quality annotations and, together with $X^{HQ}$, forming a larger training set of improved quality for the segmentation network (column HQ + upgraded). In order to ensure that the synthetic data cannot be easily segmented based on the pixel intensity levels, we use as baseline a simple thresholding solution in which the input images are segmented by selecting a threshold via grid search with a step of 1% of the maximum pixel intensity. For each data set, we select a single threshold that yields the highest Dice score on the training set. The low baseline results in Table 1 reflect the complexity of the simulated data sets. Our results show that, for most cases where $u_{\theta }$ improved the quality of annotations, the addition of samples with upgraded annotations translated into a higher segmentation performance of the final segmentation network on the test data.

From Table 1, we observed that adding the upgraded annotations to the training set results in better segmentation. However, this performance gain resulted from upgrading a large number of low-quality annotations, which may also prove difficult to produce in practice. To account for this, we perform an experiment analyzing the trade-off between annotation cost and performance. For a fixed number of slices, we select 10% of them to have high-quality annotations, while the rest have low-quality annotations. We apply our framework to this set of slices and compare against segmentation networks trained with low-quality annotations, i.e., 0% high-quality slices, and against segmentation networks trained on high-quality slices only, i.e., 100% high-quality slices. We define the annotation cost as the *equivalent number of low-quality annotations* that would be produced with the same effort as a given annotation. For instance, for a low-quality annotation, the equivalent number of low-quality annotations is 1, while for a high-quality annotation, this number will differ depending on the particularities of the task, such as the data sets or the experience of the annotators, as is the case with works comparing annotation costs in the literature [41,42]. For illustration purposes, we consider the equivalent number of low-quality annotations for a high-quality annotation to be 5. We observe in Figure 6 that, except for bias perturbation, the segmentation networks trained with our framework are the most cost-effective option for reaching the highest Dice scores. When it comes to bias, the variation in cell size induced in the training set with low-quality annotations forces the network to learn an “average” cell size that matches more closely the ground truth in the test set. However, in cases where the bias is more systematic, we expect a considerable drop in performance for the networks trained only with low-quality labels.

### 3.3. Enhancing Manual Annotations

In Section 3.1, we showed that the upgrade network is able to improve low-quality annotations of synthetic images under various circumstances. Here, we expand our analysis by validating our observations on real cell images. We integrate the two described real data sets in a scenario emulating the process experts may undertake to enhance the quality of their annotations. Our goal is to assess whether the quality gains reported in Table 1 can be similarly reproduced on real manually-annotated data. We consider a setup where the constraints on the annotation process are accurately captured by the perturbation functions used during the training of the upgrade network. With omission, we model an expert that deliberately ignores most cells in an image, focusing only on 30% of them. Inclusion allows for the presence of other structures that, for instance, can result from using networks trained on other cell data sets, or from foundation models. Bias would allow the annotator to either focus on the “core” of the cell, as shown in Figure 7m,o, or on the wider cell area without rigorously delineating the boundaries. Figure 7 shows the results of the upgrade network trained on 24% of the training samples of EPFL, and on 20% of Lizard’s, respectively. We notice, both qualitatively and quantitatively, that our solution can successfully upgrade annotations affected by high perturbation levels, requiring a relatively low number of high-quality annotations for real, more complex data sets. Also, the large quality increase for omission and bias highlights the potential of our framework to expand the size of cell data sets with relatively low effort for producing the low-quality annotations.

### 3.4. Case Study: Upgrading Low-Quality Predictions

We showcase here an example where the upgrade network can be applied in a scenario requiring no manual annotation cost for producing the low-quality annotations. In this case, $X^{HQ}$ can be employed to train a segmentation network whose predictions can then be further used as the cheap annotations of $X^{LQ}$. We consider the predictions of a segmentation network trained with 10 well-annotated samples of Lizard data set in a setup similar to [18]. We use the same $X^{HQ}$ for training our upgrade network. We opted for a set of perturbations that would guide $u_{\theta }$ to compensate for prediction inaccuracies that we visually assessed. At each training iteration, we perform a 50% omission, followed by the inclusion of 10% of the segments extracted by Felzenszwalb’s algorithm [43] to emulate missing or mispredicted structures. We add salt and pepper noise with a 10% probability to mimic the observed gaps in the segmented area as well as the small clusters of false positive pixels that can be noticed in Figure 8d. Finally, the resulting label is subjected to bias perturbation with a bias of 6 to guide the upgrade network towards better delineation of cell boundaries. The results, shown in Figure 8, demonstrate the potential of our method to refine the predictions of an undertrained segmentation network. We achieve a 22% improvement in the quality of the predictions without requiring additional supervision. Moreover, by visually inspecting the results, we notice that $u_{\theta }$ achieves good separation between the individual shapes, a property not captured by the Dice score metric. These delineated shapes can then be used, for instance, to facilitate a further instance segmentation step.

## 4. Discussion

Our results reported in Table 1 indicate that, with as few as 10 well-annotated images, we can improve low-quality annotations to a level comparable with the gold standard. In addition, as can be seen in Figure 4, the performance of the upgrade network relative to the size of the high-quality data set follows a logarithmic trend. Therefore, continuously increasing the size of $X^{HQ}$ will not generate meaningful improvements. By knowing the logarithmic trend of the performance of $u_{\theta }$, the end user of our framework would benefit from being able to decide more easily when enough high-quality data has been gathered and annotated, since, once $u_{\theta }$ performs well for a certain size of $X^{HQ}$, little improvement can be expected when the size of the training is increased. Furthermore, we showed that the upgrade network produced positive results for all considered cell data sets. The only requirements are a small high-quality set of annotations, a separate larger set of low-quality annotations, and a perturbation function that can map a high-quality annotation to multiple lower-quality versions of it, resembling the quality within the low-quality set. Since our requirements are independent of the data set, we expect our method to also work on other image modalities where our assumptions are met. This also applies to data collected in the three-dimensional regime, such as tomography. In this case, our framework can be applied on each individual slice separately.

We observed that using both the upgraded annotations of the low-quality set together with the small well-annotated set generally results in higher segmentation scores. Moreover, we noticed that the highest Dice scores are obtained when the upgrade model is both trained with and applied to annotations perturbed with 70% omission, 70% inclusion, or a bias of 6. We also saw in Figure 6 that our framework can be a cost-effective solution to increase the performance of segmentation networks when the annotation time is a constraint. Moreover, by comparing with other works targeting the enhancement of imperfect annotations, we showed that our upgrade network can handle a wider variety of perturbations than existing techniques. Thus, our solution is well-suited for being embedded into an annotation process with limited resources, rather than for fine-tuning, where there is a wide gap between the cost of producing a low-quality annotation and the cost of producing a high-quality one. For instance, for automatically-produced annotations by a non-learning algorithm, the only costly requirement would be to manually enhance a small proportion of them, on which the upgrade network can be trained. Moreover, as shown in Figure 8, our solution is flexible enough to be used for upgrading predictions of a network trained with insufficient data. These upgraded predictions can then be used to enlarge the existing data set or be further adjusted by experts, reducing the overall annotation time.

We also noticed the benefit of training for high perturbation levels, i.e., 70% omission, 70% inclusion, and a bias of 6, when we tested the robustness of our solution with respect to discrepancies between the perturbation levels used to train the upgrade network and the perturbation levels in the low-quality set. In Table 2, we saw that, generally, when we train for the highest perturbation level we reach comparable, or higher, performance than when training on the same perturbation applied to generate the low-quality set. Since, in practice, the annotation inaccuracies can have a systematic, i.e., annotator-specific, component and a random component, it may prove impossible to exactly model these inaccuracies through perturbations. Thus, the robustness to discrepancies in perturbation levels shown by our framework can indicate its potential applicability in practical scenarios. We additionally showed that our framework is robust to reductions in the quality of $X^{HQ}$. Figure 5 shows that we can expect a relatively small drop in performance when we moderately reduce the quality of the well-annotated set. This observation may imply that the annotation process of $X^{HQ}$ can become less costly, e.g., requiring fewer experts per high-quality annotation, while still being able to produce annotations to train a well-performing upgrade network. However, the less information is present in an annotation, e.g., small cell areas, the more sensitive the framework becomes to inconsistencies.

Given that we focused solely on cell segmentation, we are unable to conclude with certainty whether or not our framework is applicable to other image segmentation applications where the goal would diverge from the cell segmentation setup, for instance by requiring the segmentation of a single contiguous target object. However, considering that our framework does not demand a specific type of annotation, as long as sufficient realistic low-quality versions of the high-quality annotations can be created with enough variety between them, we expect the upgrade network to still be applicable. Despite this, further experimentation is required to ensure that our requirements are met by other segmentation applications. Another limitation presented by our work is the lack of integration of the third dimension for volumetric data sets. This can be tackled in the future by, for instance, employing an architecture with 3D convolutions as the upgrade network. Finally, throughout our experimentation, we upgraded only annotations suffering from high levels of inconsistencies, while ignoring the fine-tuning of less severe cases. We expect our upgrade network to not perform similarly well on such cases, given that the small errors would not allow for much variation in the generation of the low-quality versions of the annotations. This would then impede the network from learning a generalizable mapping from a low-quality annotation to a high-quality one.

## 5. Conclusions

We presented our framework for enlarging training data sets with limited human annotation costs by only requiring a small set of data with high-quality annotations and a larger set with low-quality annotations that would require little or no human annotation effort. We utilize a small high-quality data set whose annotation quality is reduced for providing it as input to an upgrade network that learns the mapping from a low-quality annotation to a high-quality one. We then use the upgrade network to enhance the annotation quality of the larger low-quality set.

We observed that our solution is applicable to at least three types of annotation inconsistencies (omission, inclusion, and bias), that it is robust to changes in the annotation quality of the training set, and that it can have wider applicability than existing works. We showed that our work can be applied to enhance the low-quality predictions of a network trained on an insufficient number of samples. Finally, we showed that the networks trained on data sets enlarged by our method present higher segmentation scores than only training on high-quality data.

## Figures and Tables

**Figure 1 jimaging-10-00172-f001:**
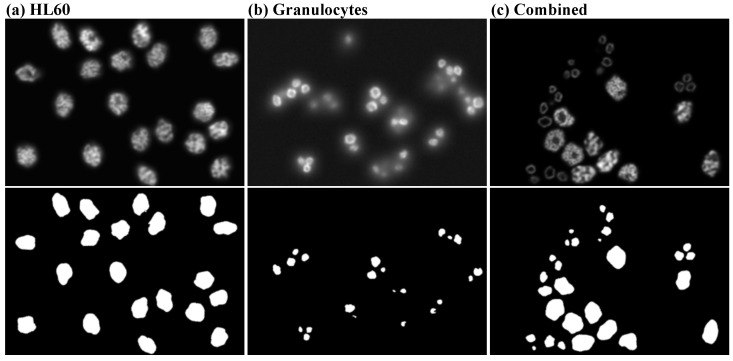
Sample slices and their corresponding high-quality annotations for the synthetic data sets we considered for analysis. The slices are produced with a virtual microscope [29].

**Figure 2 jimaging-10-00172-f002:**
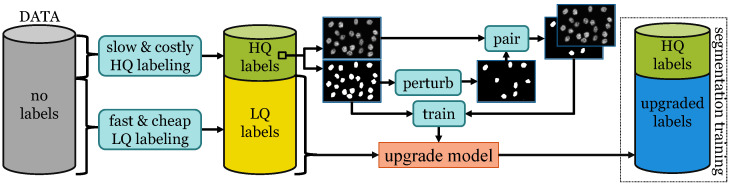
Workflow. We train the upgrade model on a small set with high-quality labels. We apply the trained model to upgrade the low-quality labels of a larger set. We enlarge the initial high-quality set with the upgraded labels and we use the combined set for segmentation training.

**Figure 3 jimaging-10-00172-f003:**
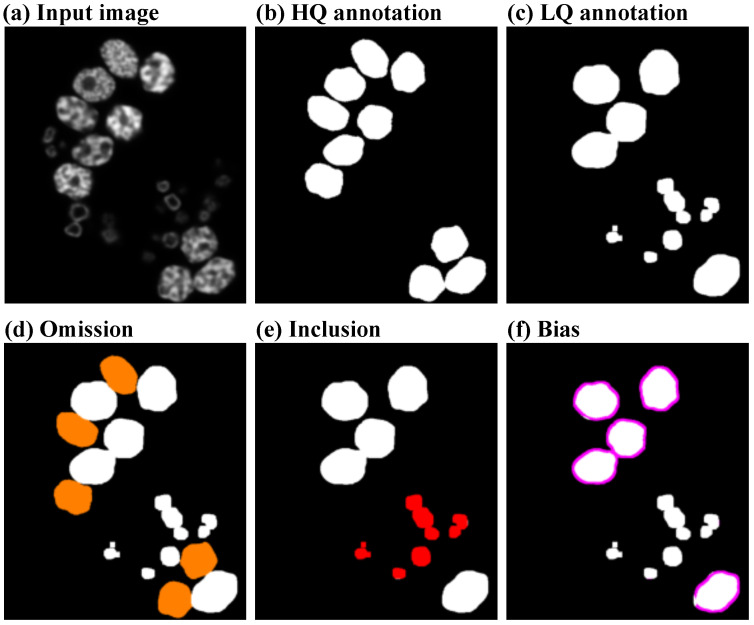
An example of the perturbations applied to the high-quality annotations. Figure (**a**) presents an input image from the combined data set, where the HL60 cells are the target cells and the granulocytes are the included cells. The high-quality annotation corresponding to the input is shown in (**b**). The low-quality version of the annotation shown in (**c**) is affected by 50% omission, 50% inclusion, and a bias of 6. The omission perturbation is represented by the orange omitted cells in Figure (**d**), inclusion by the red shapes in (**e**), and bias by the magenta contours in (**f**).

**Figure 4 jimaging-10-00172-f004:**
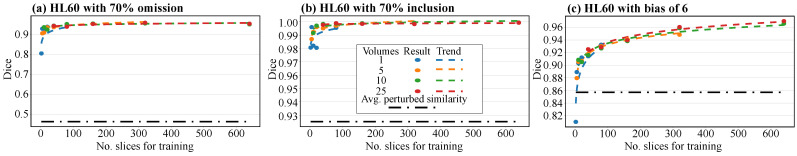
The Dice similarity with the ground truth test set of the upgraded annotations as a function of the total number of slices used for training and the number of volumes from which the slices were selected for the HL60 cells (represented using a different color). The colored dots represent the experimental results, while the colored dashed lines are showing the general trend of the results. The straight dash-dotted line represents the average Dice similarity with the ground truth test set of the perturbed annotations before the upgrade.

**Figure 5 jimaging-10-00172-f005:**
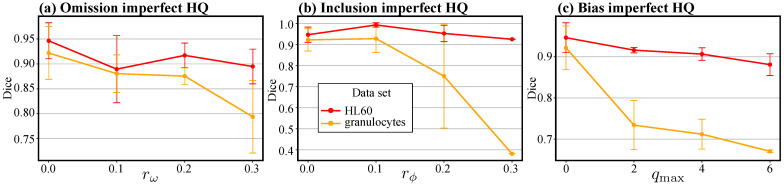
The Dice similarity with the ground truth test set of the upgraded annotations as a function of the perturbation level present in the high-quality training set of the upgrade network. The vertical bars correspond to the standard deviation of the results.

**Figure 6 jimaging-10-00172-f006:**
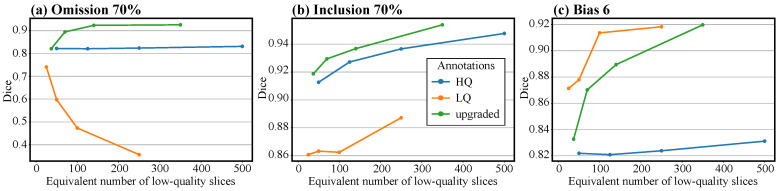
The Dice similarity with the ground truth test set of the segmentation networks as a function of annotation cost. The results in Figure (**a**) correspond to the HL60 data set where the low-quality annotations suffered from 70% omission, the results in Figure (**b**) correspond to the combined data set with HL60 cells as targets and 70% inclusion in the low-quality annotations, and the results in Figure (**c**) correspond to the HL60 data set with a bias of 6 in the low-quality annotations.

**Figure 7 jimaging-10-00172-f007:**
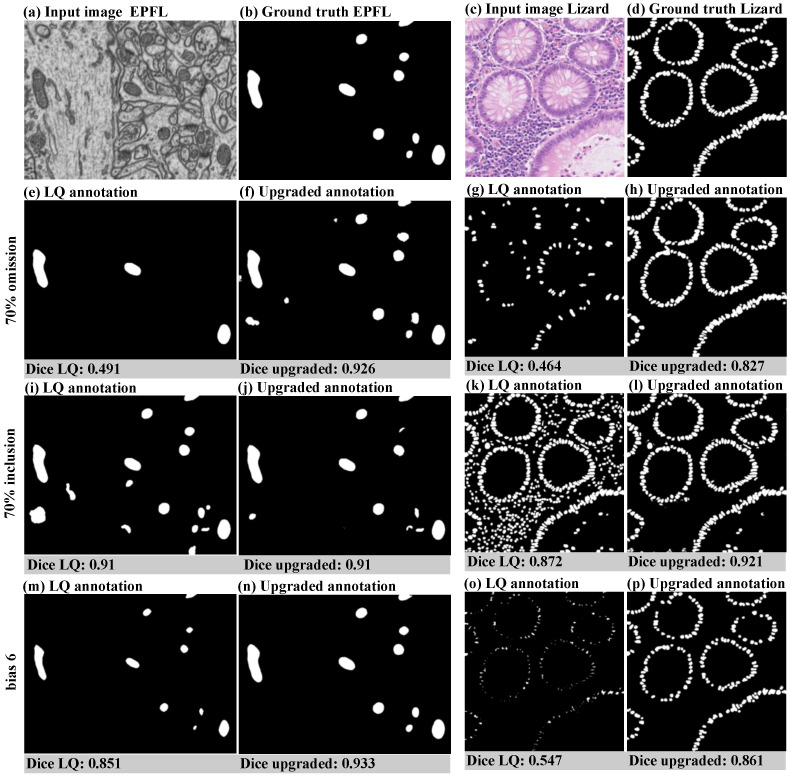
An example of perturbations applied to the real data sets paired with upgraded annotations. Figures (**a**–**d**) show the input image paired with its corresponding ground truth for EPFL and Lizard. Figures (**e**–**p**) present the perturbed-upgraded annotation pairs for 70% omission, 70% inclusion, and a bias of 6, respectively. The results below the images represent the Dice similarity between the ground truth, the low-quality annotations, and the upgraded annotations, respectively. Both metrics were computed on the entire test set.

**Figure 8 jimaging-10-00172-f008:**
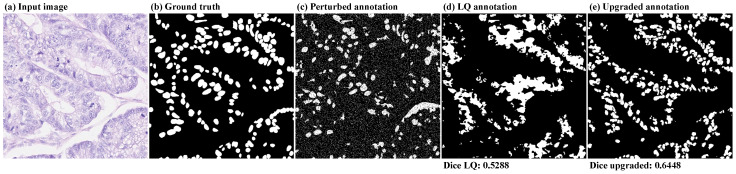
An example of an application where our framework can upgrade the predictions of a segmentation network trained on insufficient data. Figures (**a**,**b**) show the Lizard input image together with the corresponding ground truth. Figure (**c**) is an example of a perturbed annotation used during the training of the upgrade network. Figure (**d**) shows a low-quality annotation produced by the segmentation network with its upgraded version presented in Figure (**e**). The results below the images represent the Dice similarity between the ground truth, the low-quality annotations, and the upgraded annotations, respectively. Both metrics were computed on the entire test set.

**Table 1 jimaging-10-00172-t001:** The Dice similarity with the ground truth test set of the annotations affected by perturbation and the upgraded annotations, as well as of the predictions produced by segmentation networks trained only on the high-quality data, on the high-quality data together with the data with upgraded annotations, and the results of using thresholding as baseline. In each row, the largest value is highlighted in bold. The training setup indicates the data set on which the upgrade network was trained, as well as the total number of slices used for training and the number of volumes from which the slices were selected. The cell types marked with an asterisk were obtained from the combined synthetic data set.

	Training Setup for Upgrade Network			Quality of Training Annotations		Quality of Segmentation Network				
						Training Data				
Perturbation	Data	Vols.	Slices	LQ	Upg.	HQ	HQ + upg.	HQ + LQ	LQ only	Thrs.
70% omission	HL60	10	10	0.462	**0.939**	0.823	**0.929**	0.311	0.311	0.887
	gran.	10	80	0.495	**0.92**	0.892	**0.894**	0.41	0.414	0.732
70% inclusion	HL60 *	10	10	0.925	**0.992**	0.913	**0.962**	0.891	0.89	0.892
	gran. *	10	10	0.381	**0.98**	0.856	**0.898**	0.364	0.353	0.214
bias 6	HL60	10	10	0.857	**0.909**	0.823	0.923	0.931	**0.933**	0.887
	gran.	10	40	0.675	**0.865**	0.868	**0.877**	0.827	0.81	0.732
30% om. 30% inc. bias 4	HL60 *	10	10	0.71	**0.929**	0.913	**0.934**	0.739	0.745	0.892
	gran. *	10	10	0.54	**0.86**	**0.856**	0.854	0.505	0.5	0.214

**Table 2 jimaging-10-00172-t002:** The Dice similarity with the ground truth test set of the upgraded network trained on various degrees of perturbations. The perturbations present in the low-quality set are 30% omission, 30% inclusion, and a bias severity of 4, respectively. For each perturbation type, the highest score is highlighted in bold.

	Training Perturbation for Upgrade Network								
	Omission			Inclusion			Bias		
	20%	30%	50%	20%	30%	50%	2	4	6
HL60	0.955	**0.972**	0.952	0.973	0.972	**0.986**	0.915	0.918	**0.926**
gran.	0.838	0.86	**0.93**	**0.984**	0.98	0.981	0.821	0.837	**0.884**

**Table 3 jimaging-10-00172-t003:** The Dice similarity with the ground truth test set of the predictions produced by three different segmentation networks designed for training with incomplete/noisy annotations. The models were trained on 10 volumes of the combined set of HL60 and granulocytes with the HL60 cells as the target cells. The upgrade network of our method was trained using 20 well-annotated slices, which were also included in the training set of the other two methods. For each perturbation, the highest score is highlighted in bold.

	Perturbation		
Method	70% Omission	70% Inclusion	Bias 6
Partial Labeling [10]	0.906	0.859	0.803
Confident Learning [13]	0.381	0.888	**0.947**
Ours	**0.923**	**0.962**	0.916

## Data Availability

The synthetic data sets can be downloaded from https://cbia.fi.muni.cz/datasets/ (accessed on 15 June 2023), EPFL data can be found at https://www.epfl.ch/labs/cvlab/data/data-em/ (accessed on 10 April 2023), and Lizard is available at https://www.kaggle.com/datasets/aadimator/lizard-dataset/data (accessed on 10 April 2023). The code used to perform the experiments can be found at https://github.com/SerbanVadineanu/upgrading-annotations.git (accessed on 15 June 2023).

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
