# Peer review of "Reducing Manual Annotation Costs for Cell Segmentation by Upgrading Low-Quality Annotations†"

_2313-433X, 2024, doi:10.3390/jimaging10070172_

Round 1
Reviewer 1 Report
Comments and Suggestions for Authors
This manuscript aims to reduce the complexity of creating high-quality annotations set by upgrading lower-quality annotations, it might be a useful method in deep learning algorithms for some scenarios which are difficult to collect large data sets. The authors show an effective method to upgrade annotations with small well-annotated set, and increase the quality of the predictions of image segmentation. Some suggestions were stated bellow:
If there is a principle of low quality annotations producing, such as a distribution of the factor q of bias perturbation, or the range of omission rate and inclusion rate?
In section 2.3.3, the image was eroded or dilated by structuring element (se), the size of the se should be stated in the paper. And does it make a difference on the result?
In fig.5, the Dice of HL60 is better than granulocytes, the difference can be maximized as 0.5 in inclusion What are the reasons for that ?
Reviewer 2 Report
Comments and Suggestions for Authors
This manuscript presents a deep learning method for upgrading low-quality labels to high-quality labels for cell image segmentation. This idea is interesting. However, the following concerns need to be addressed.
1. The definitions of ‘low-quality’ and ‘high-quality’ are quite subjective, although I understand what the authors mean for each one. Is there any objective metric to grade the quality of a label? I would appreciate that the authors can develop such metrics. That would be much greater contributions.
2. In producing ‘low-quality’ labels, I would suggest using non-deep-learning techniques such as graph cuts. They can produce promising segmentation results that may be close to the so-called ‘high-quality’ labels. In this era of deep learning, I think the conventional image processing techniques should not be abandoned.
3. The network structure should be clearly described in Section 2. Materials and Methods.
4. Part of the Results section should be moved to the Materials and Methods section, such as the descriptions on the experimental setup and on the datasets.
5. Please consider revising the caption of Section ‘2.1. Setup’.
6. More descriptions on the hyperparameters should be described, including, but not limited to the learning rate.
7. ‘This paper is an extended version of our paper published in [1].’ This sentence should be moved to the Introduction section. It may be appropriate to be located at the end of Introduction. Moreover, please describe the specific extensions compared to [1]. That is, what is the difference between this manuscript and [1]?
8. Please clarify the limitations of this study at the end of Discussion.
9. Can the methods described this work be extended to other image segmentation applications? What may be the considerations? Please discuss these in Discussion.
Reviewer 3 Report
Comments and Suggestions for Authors
Vadineanu et al. describe an innovative method for performing cell image segmentation using partially annotated labels or low-quality annotations. The authors demonstrate that in their workflow, they use a small set of high-quality annotations to train a convolutional neural network to obtain a model. This model is then used to train on a large set of low-quality annotations, which can reflect inconsistencies and errors in the segmentation. By combining the original high-quality annotations with a larger set of low-quality annotations, the method perturbs the original annotations, incorporating the types of errors present in the original segmentation dataset. This process upgrades the original segmentation datasets by integrating information derived from the errors into a single and improved segmentation dataset.
The manuscript is well written and the authors provide a detailed explanation of the process. The results are extensively discussed to highlight the strengths and drawbacks of the technique. Therefore, I recommend the manuscript for publication after some minor adjustments.
1) Although the examples illustrate how the method works and can inform segmentation for multiple microscopic techniques, the authors do not clarify how this would help with data collected in the three-dimensional regime. For example, can the authors provide comments on how their method would assist with tomographic data?
2) In addition, can the authors comment on what the limitations of the method when applied to different imaging techniques would be? For example, would such an approach provide the same degree of updated information in light and electron microscopy? And how would it compare to the information provided by nano-CT data?
Round 2
Reviewer 2 Report
Comments and Suggestions for Authors
The authors have made a nice revision. Thanks. All of my concerns have been addressed.